# The Use of Drones in the Spatial Social Sciences

**Ola Hall *** and **Ibrahim Wahab**

Department of Human Geography, Lund University, SE-223 62 Lund, Sweden; ibrahim.wahab@keg.lu.se
* Correspondence: ola.hall@keg.lu.se; Tel.: +46-73-374-7849

**Abstract:** Drones are increasingly becoming a ubiquitous feature of society. They are being used for a multiplicity of applications for military, leisure, economic, and academic purposes. Their application in academia, especially as social science research tools, has seen a sharp uptake in the last decade. This has been possible due, largely, to significant developments in computerization and miniaturization, which have culminated in safer, cheaper, lighter, and thus more accessible drones for social scientists. Despite their increasingly widespread use, there has not been an adequate reflection on their use in the spatial social sciences. There is need for a deeper reflection on their application in these fields of study. Should the drone even be considered a tool in the toolbox of the social scientist? In which fields is it most relevant? Should it be taught as a course in the social sciences much in the same way that spatially-oriented software packages have become mainstream in institutions of higher learning? What are the ethical implications of its application in spatial social science? This paper is a brief reflection on these questions. We contend that drones are a neutral tool which can be good and evil. They have actual and potentially wide applicability in academia but can be a tool through which breaches in ethics can be occasioned given their unique abilities to capture data from vantage perspectives. Researchers therefore need to be circumspect in how they deploy this powerful tool which is increasingly becoming mainstream in the social sciences.

**Keywords:** drones; legislation; ethics; spatial social sciences

## 1. Introduction

The term drone (For the purposes of this paper, whenever we use the term drones, we are referring to unmanned aerial systems in their use to capture photographic data, and as this is used in the social sciences. Our focus therefore is the photographic product that can be collected from drones and not the devices themselves), originally referring to the male bee, is the everyday name for autonomous aircraft. Drones tend to evoke memories of warfare as they were first flown during the First World War, during which they were often launched using a catapult. Since then, they have been used as tools for reconnaissance, for deploying propaganda leaflets, as decoys for missile launches, or even as actual combat platforms in multiple theaters of war. In more recent times, drones have become household names for their non-military uses. There have been several attempts to delink them from their militaristic past. These efforts include popularizing more civilian-leaning names for these systems. To this end, common monikers in academic literature include unmanned aerial vehicles (UAV), remotely-piloted aircraft (RPA) or vehicle (RPV), unmanned aerial system (UAS), or the recent, gender-neutral form, uncrewed aerial vehicle (also UAV). Notwithstanding these efforts, the traditional terminology has, however, stuck, and so even now, major industry players offer 'drones' and 'mini-drones' as flagship products.

Within the last decade, drones have become a much more common feature of life. This is largely attributable to the fact that they have become relatively cheaper to manufacture compared to just two decades ago. This is mainly due to significant technological advances in computerization and miniaturization. The former has exponentially increased the processing power of computers and researchers' ability to process data from drones, even on low-cost laptops, while the latter has dramatically reduced the cost of production with

less expensive components such as carbon fiber. Increasing civilian use of drones has also accompanied improvements in features aimed at augmenting safety. These features include obstacle-avoidance and vertical take-off and land (VTOL) systems. The latter allows take-off and landing of the drone system even in challenging terrains, while the former helps prevent mid-flight collision with other aircraft, trees, or buildings.

The application of drones is expanding just as their use in multiple facets of life is growing. In addition to being used as leisure tools, drones have several applications in weather forecasting, search and rescue operations, disaster management, crowd control, and delivery of vaccines and blood for transfusion, among others. One area that has also begun seeing the increased application of drones in the last decade is academia. In this regard, both the physical as well as the social sciences have found it to be a useful tool. Some disciplines, by virtue of their subject matter and focus, find the incorporation of drones into their research agenda easier than others. For instance, in the physical sciences, the subject matter is often physical, and the unit of analysis could be rocks, stars, plants, or animals. Studying such using drones is easier and more straightforward than the social sciences, in which the subject matter is society itself and the unit of analysis is often humans. The latter are more complex units of study because they are sentient, conscious, and can modify their activities under study. Related to this are ethical dilemmas as well as questions about objectivity when we study our, as well as other, societies. Given our biases, both declared and undeclared, as well as recognized and unrecognized, there is often the need for greater reflexivity in social science studies, especially in collecting data.

Generally, social scientists employ a range of tools and methods to collect data for their studies. These methods range from experiments, surveys, interviews, focus group discussions (FGDs), participant observations, life histories, and documentation analysis, among others. While each of these methods have their strengths and weaknesses, their appropriateness depends on the context of studies. Some social science disciplines are more inclined to relying on visual methodologies than others. The social science disciplines of anthropology, archaeology, economics, geography, history, and sociology have found drones to be useful research tools. Of these, anthropology and geography are unique in their reliance on the visual and visual images to construct their knowledge [1]. Given drones' advantage of providing access to a bird's eye view of the geographical space [2], and geography's preoccupation as a spatial science, drones have found a much more accepting audience among geographers compared to other social scientists. Even within geography, not only are drones helping to bridge the gap, but they are also offering new opportunities for collaborative research between human and physical geographers given that these two subdisciplines often approach the application of drones differently [3].

In this conceptual paper, we aim to discuss the use of drones as social science research tools. In this vein, we focus the discussion on three main themes: teaching their use as a course in the social science faculties of universities, legislations governing their use across countries, and the ethical and political hurdles that need reflection in their application, particularly in the social sciences. While the civilian use of drones for surveillance and policing to fight crime is generally socially acceptable [4], there is often a certain level of uneasiness and strong pushback against a continuous, universal, and an all-seeing flying big brother in the sky [5], given the potential for abuse and concerns for privacy. These reflections are critical as drones continue to become mainstream tools in the toolbox of the social scientist.

## 2. Drones as Social Science Research Tools

The application of drones in the social sciences as data collection tools comes on the back of the use of satellite imagery in the same endeavours. The latter can be traced to the mid-1990s when the National Aeronautics and Space Administration (NASA) approached the research community to realize the potential of satellite imagery to specifically address questions which social scientists are preoccupied with. Notwithstanding the high expectations from this collaboration expressed in People and Pixels: Linking Remote Sensing and

Social Science [6], the results have been meagre, and their added value questioned. Much of the difficulties that limited the success of using remotely sensed satellite imagery-coarse resolution of most readily available datasets, the challenge with cloud cover, particularly in the tropics, and limitations relating to temporal resolutions have persisted. This is despite the significant strides that have been made in this area in the last two decades. It is on the back of these challenges that other platforms have been proposed as alternatives to satellites as remote sensing platforms for collecting critical data about the earth's surface. As the third generation remote sensing platform-with piloted aircraft as the first generation and earth-orbiting satellites as the second generation [7], drones are proving much more ubiquitous in terms of their application in scientific research.

There are, of course, substantial differences between drone data and satellite imagery and, as such, the two are not comparable. An important differentiating factor is the scale of application. While satellites are ideal when the macro view of the terrain is needed, due to the larger spatial coverage, drone imagery is better suited for a micro view of the landscape, given its higher-centimetre-level-resolution. Some studies have catalogued the pros and cons of each platform and shown where each performs optimally [8,9], others have been preoccupied with integrating them in a synergistic manner [10,11]. The general trend, with regard to spatial resolution, is a continuous increase, with some satellite platforms now offering sub-meter resolutions. This opens the door for greater applications that were hitherto virtually impossible. The recent use of the 1 m resolution Terra Bella satellite imagery for measuring smallholder productivity in Western Kenya is a case in point [12]. Thus, each platform and the resultant data they generate meet specific needs. In some cases, however, drone data can be up-scaled to cover larger areas. This of course, raises questions of cost-effectiveness [13].

Drones and drone imagery position the researcher and the objects of interest closer together, both physically as well as conceptually. Compared to satellite imagery, the low flying altitude, small area coverage, and the detailed visuals that drones offer create a familiar perspective, closely related to traditional field work. Unlike satellite imagery, drone imagery is usually ready to be used as a map base or in photo elicitation interviews [14–16]. They can also be processed, classified, and analyzed in a more conventional remote sensing way [17,18]. Thus, drone imagery can either be used on its merit or to improve the quality of data that other more conventional methods of data collection in the social sciences have produced. For instance, van Auken, Frisvoll [19] enumerate the advantages that photo-elicitation interviews have over more traditional social science research tools, such as the provision of tangible stimuli for more effectively tapping into informants' tacit, and often unconscious knowledge, consumption of representations, images and metaphors, and thus leading to the production of different and richer information than other techniques, while also helping to reduce differences in power, class, and knowledge between researcher and researched. In the developing world, which is invariably the 'data-poor' world, drone imagery has proven to be an indispensable tool for research.

The geographical applications of drones are perhaps more widespread than in other fields. In the sub-field of physical geography, drones have gained wide acceptance for studying rock weathering [20,21], for river bed monitoring [22,23], and for restoration [24]. In this area, studies have progressed beyond proof-of-concepts to real-world applications for geomorphological change detection and mapping, vegetation mapping, habitat classification and sediment transport path delineation [25]. Further downstream, Callow, May [26] used a drone to generate high-accuracy, centimeter-resolution digital topographic models which offer insights into the likely consequences of inundation and the dynamics that control low-gradient sedimentary landforms. It is not surprising that geography in general, and the subdiscipline of physical geography in particular, were always going to be more accepting of drones due mainly to the their 'vertical' and 'visual' character. The proliferation of drones as research tools, however, avails further opportunities for intra-discipline collaboration between physical and human geographers [3].

In agricultural geography, drone applications include mapping crop condition and yield estimation [18,27,28], crop classification [17], seedling emergence assessment, crop damage assessment, weed detection, and mapping [29]. In general terms, drones have been heralded as the right tools for making agriculture smarter, especially in Sub-Saharan Africa where the application of the first- and second-generation remote sensing platforms have met with largely limited success. This limited success is due to such factors as costs of acquisitions, cloud cover, and low spatial and temporal resolutions vis-à-vis the predominance of small farms in SSA. Multiple reviews such as those by Daponte, De Vito [30], Puri, Nayyar [31] have chronicled the use of drones in the field of smart agriculture. Iost Filho, Heldens [32] more specifically reviewed the application of drones as noninvasive crop monitoring systems in precision pest management. Three main niches exist in this subfield for drone application: (1) scouting for problems; (2) monitoring crops to prevent/reduce losses; and (3) planning crop management operations [33]. Similarly, Barbedo [34] offers a more comprehensive and critical review of the use of drones in this area, chronicling the major milestones, the main research gaps and possibilities for future research with the application of even newer techniques of machine learning on drone image analysis.

This is, however, not to assert that drones are not already useful research tools in the other subdisciplines of human geography. In tourism studies, for example, the usefulness of drones continues to grow. Here, drones are being used for monitoring and patrolling tourism activities for safety and security as well as for virtual tourism systems [35]. For virtual tourism, drones serve as destination marketing tools to produce large amounts of visually appealing footages of various destinations [36]. On the part of tourism service providers, major considerations regarding economic viability and operational feasibility need to be addressed in order to efficiently deploy drones in the tourism sector [37]. While tourists tend to have a better appreciation of the potential uses of drones compared to managers of tourist centers, there is need to set boundaries of what is acceptable [38]. Beyond tourism studies, drones are finding applications as research tools in cultural geography, health geography, rural geography, transportation geography, and urban geography, among others.

In the area of environmental geography, for instance, community drones for natural resource management and conservation is a strongly growing niche [39–42]. Much of the work in this area emanates from Latin America and, to a lesser extent, South-East Asia. Drone applications are most useful in cases where the study locations are usually difficult to physically access. A number of reviews by those such as Paneque-Gálvez, Vargas-Ramírez [43], Canal and Negro [44], Beaver, Baldwin [45], have emerged in this area that point to current and potential significant contributions that drones can be put to in natural resource management. Cummings, Cummings [46] go a step further to demonstrate how drones can be adapted in indigenous peoples' dominated settings in a collective and concerted manner. Following such a collaborative approach can help build mutually beneficial relationships, as it respects indigenous culture and customary norms, which in turn augurs well for a sustainable monitoring and protection of natural ecosystems. Vargas-Ramírez and Paneque-Gálvez [39] provide a broad overview of this growing field of community drones, finding that local knowledge is often neglected or undervalued, and emphasizing the need to recognize indigenous peoples' territorial rights. Done well, participatory action mapping (PAM) using drones can be useful for bolstering the political and legal claims of indigenous communities to counteract land grabs by foreign entities [47,48]. Conversely, unintended negative consequences of PAM include fragmentation and conflicts among indigenous communities and the facilitation of land acquisitions, either by the state or corporations, following legal recognitions [49]. There is also the need for researchers engaged in PAM to pay attention to the digital divide that often exists between them and indigenous communities, a situation which is symptomatic of broader socioeconomic and political inequalities which are largely legacies of colonialism [50].

In archaeology, drones are becoming increasingly useful in studying previously unrecognizable features. For example, Cucchiaro, Fallu [51] demonstrate that orthomosaics from drones provide an accurate and high level of detail of the terrace landscape, the archaeological features and sediment stratigraphy along an excavation trench previously unobserved. Similarly, Brown, Walsh [52] show the beneficial use of drones to map multi-faceted terraces under intensification and diversification. In landscape archaeology, Stek [53] demonstrates the utility of drones for detecting previously undocumented subsurface archaeological artifacts in mountainous, Mediterranean landscapes. Campana [54] provides an excellent review of the application of drones in archaeology and delineates five main areas of application: exploratory aerial surveys, survey of archaeological sites and landscapes, three-dimensional (3D) documentation of excavations, 3D surveys of monuments and historic buildings, and archaeological surveys of woodland areas. Just like in tourism studies, the application of drones, which makes hitherto unobservable sites accessible, also requires some safeguards to ensure that aerial photos do not contribute to looting and destruction of heritage sites [55].

Different fields incorporate drones into the fieldwork and studies to varying extents. The broad field of geography is, however, relatively more predisposed to employing drones compared to other fields because it uses the full sweep of quantitative and qualitative methods and places greater emphasis on fieldwork and mapping. This is underpinned by its special focus on spatial analysis and areal differentiation. Perhaps, through more widespread teaching of drones as a stand-alone course in the social sciences in universities, other fields might come to realize their value and how they can be adopted and utilized to suit each discipline's peculiar needs.

## 3. Teaching Drones in Higher Education

The teaching of drones in institutions of higher learning is fast catching on, notwithstanding the substantial capital outlay that this involves due to the infrastructure demands that it entails. The teaching can, broadly, be categorised into two main areas: (1) teaching it as hardware, including the development of technical improvements to the drone's navigation systems; and (2) teaching drone-based data capture and processing. The first is the kind of stuff that more technical departments of universities such as engineering already do, and this is not the focus of the present paper. Here we limit ourselves to the social sciences, and thus the second, which is the teaching of the capture and processing of drone-based data.

In this era of rapid data collection, drones have emerged as a well-established geospatial technology for collecting and analysing primary remote sensing data. In terms of importance, they are poised to be as revolutionary for geography in the same order of magnitude as other spatially-oriented software packages. They offer a method for collecting and accumulating data from strategic viewpoints [56] and at such fine spatial resolutions that there are many social science disciplines that can benefit from this vantage perspective. Given their ubiquity in society and increasing applications, even in the social sciences, there is increasingly obvious need for having a dedicated course on drones in research-oriented universities. To be fair, much of what we propose here, in terms of teaching drones, is already being done by many engineering departments across many universities. Our focus here relates to flying the drones, capturing spatial data in photographic format, processing these into orthomosaics, and the application of these in the social sciences and the ethical implications that this entails. It is our position that the reflexivity and reflections that arise when social scientists undertake these processes themselves are markedly different from those that arise when the orthomosaics are presented to them for analysis. Hence the need to teach these in universities to social science students even if photogrammetry has been taught for years to engineers. Herein lies the gravamen of the argument for the teaching of drones as an important tool in the toolbox of the social scientist.

Effective teaching of the principles and applications in a field such as dynamic geographic information systems and technology in higher education is usually challenging [57]

and drones are no exception in this. This is partly down to the constant change that this niche of study is subject to. Effective teaching of drones in institutions of higher learning needs to overcome two fundamental issues: teachers need to adopt and adapt new paradigms and tools while keeping up to date with newer trends in the field, and yet also develop effective methods for transferring the new competencies to students. The teaching of drones, especially when it encompasses image acquisition, data processing and interpretation, has been shown to significantly enhance students' data processing skills while enhancing their competence in handling data quality issues [58]. Consequently, this field is usually at the cutting edge of teaching and learning approaches, with traditional approaches giving way to more modern methods.

In recent times, more traditional approaches, such as pen-and-paper in lectures and laboratory exercises, are giving way to more active learning strategies such as 'flipped classrooms' [57]. Such participatory and collaborative approaches lay good foundations among students for participatory action research and popular education approaches which ensure community participation and cultural appropriateness of the methods that are employed in data collection using drones [39]. These, however, often require further training not only on the part of students, but instructors as well. On the part of instructors, there is often the need to allocate extended periods of preparation for classes to keep abreast with software updates as well as new trends and developments in the general field of geographic information science and technology [57] and the specific field of image analysis, especially using the artificial intelligence techniques of machine learning and deep learning using big data. Holloway, Kenna [59] further posit that, with regards to new technologies, such new approaches foster teamwork, peer-to-peer learning, and positively reinforce the uptake of such technologies in fieldwork.

Already, some institutions are setting the pace in teaching drones both at undergraduate and advanced levels. In the United States, the Drone Journalism programmes at the Universities of Missouri and Nebraska-Lincoln are pacesetters in drone studies. In such programmes, students are taught not only the technical skill of flying a drone mounted with cameras to collect aerial data, but also the ethics of collecting data on people in public places and the legal, safety and regulatory frameworks across various jurisdictions and areas as well as the analysis of aerial data [56]. For instance, flying regulations are different within a 2-km radius of an airport than they are for a rural area [3]. For safety reasons, special rules also apply to flying altitudes, with about 100 metres often considered a safe height. Elsewhere in continental Europe, the Oslo School of Architecture and Design is also considered an early pioneer in the teaching of drones as a course [60]. Similarly, Lund University has an aviation school which specializes in training and certification of drone pilots and is in the process of acquiring the necessary credentials from the country's transport administration. Other research-oriented universities should be following these early innovators in this endeavour. The main challenge for instructors is to cover the three fundamentals of remote sensing, these being planning, data collections, and image analysis, while minimising logistical and practical issues associated with the actual flights [61]. A further hurdle in this is securing the necessary certifications to be able to train pilots within the existing legislation framework.

Those institutions that do not have the infrastructure or which are in jurisdictions where private drone use, even for academic purposes, is significantly restricted, could liaise with already established pilot schools to train students on the technical aspects of flying while they focus on the theoretical, philosophical, ethical, and methodological aspects of drone use in the social sciences. The onus falls on geographers, both physical and human ones, to actively engage with this technology and lead cross-disciplinary discussions on not only the processing of drone data but also the ethical implications of its use. Collaborating with specialized flight training schools thus helps to overcome barriers relating to certification and licensing.

## 4. Legislations on Drone Use across Countries

At the core of the legislations regulating the use of drones is the need to ensure safety and minimize harm in the use of drones in civilian airspace. Professional use of drones necessarily needs to be guided by a multiplicity of legislations, from the national, regional, and even local levels. Regulations often relate to the flying of the drone itself, the safe and secure management of the communication to and from the drone system, and those relating to the ethical issues arising from the acquisition, processing, and dissemination of drone imagery. Underpinning the first two is the need to prevent airspace conflict and interference with commercial airport systems. The third is concerned with preventing breaches of confidentiality, privacy and safety of people and locations of national security importance.

Much like other technological innovations, regulations for drones have been playing catch-up with the proliferation and use of the devices [62]. Different countries have reached different regulation development stages for drones, with most countries, especially in Africa, having developed their national guidelines within the last half decade. Even among OECD countries, there is substantial heterogeneity in national legal frameworks on drone regulation [63]. Regulations on the use of drones are necessary due to the potential for breaching privacy, data protection, and public peace [64]. Regulations relating to licensing and operations therefore vary significantly across countries, even though a substantial proportion (40 to 85%) of the provisions of legislations governing drone usage is often sourced from the manual of the International Civil Aviation Organization [64]. Even where regulations have been harmonized, as is the case with the European Union (new regulations came into force in January 2021), stakeholders often find them cumbersome due to administrative and bureaucratic complexities in their interpretations [65]. The regulatory field will most likely continue to be characterized by fluidity in the foreseeable future. There are a few data repositories for information on drone regulations worldwide. A useful portal for the most up-to-date information on drone regulations for various jurisdictions can be found at: https://www.droneregulations.info/index.html (accessed on 16 June 2021). Here, one can access the specific website of respective national authorities responsible for licensing and issuing guidelines and regulations for drone pilots. The portal thus serves as a one-stop-shop for the most updated laws on the use of drones on each country.

A major challenge relating to the regulations is the restrictions they tend to come with. This is particularly true when regulatory agencies fear that lives could be at risk. In such circumstances, there is a tendency for broad restrictions which limit adoption and use of drones even for academic research [64,65]. These barriers are sometimes purely financial. For example, Kenya Airways has an entry-cost of about USD 1600 for a month-long course to obtain a license to fly a drone in Kenya [66]. This excludes other charges such as the cost of medical examination. The initial license issuance costs some USD 720 and this is renewable at a fee of about USD 460 [67]. This area is, however, in constant flux. Countries are regularly reviewing regulations to improve the ease of use of drones in their jurisdictions. In the United States, which is a pioneer in this area, drone operators are no longer required to pass a medical examination nor have liability insurance, for example. Drone pilots are only required to pass an aeronautical knowledge test rather than acquire any form of pilots' license.

Other portals exist to check drone operations and report incidents involving drones. The most popular of these is the drone incidents and intelligence system https://www.drone-detectives.com/ (accessed on 7 July 2021). The primary purpose of this portal is to safeguard public safety by allowing private individuals to report dangerous drone activities and to file accident incidents involving drones. Some of the details one can report include date and time of incident, the type/model of drone involved, and the altitude at which the drone was flying, as well as the specifics of the incident such as the proximity to airport airspace or military installations. Such reports are useful for regulatory institutions in their monitoring activities. Apart from showing incidents involving drones, Drone Detectives

is also useful in noting the various no fly zones in all countries. These are usually over military installations, airport airspace and public spaces such as parks and stadia.

Thus, while the fundamental role of these national regulations relating to drone use is to ensure public safety and security, some of the rules will have to be relaxed as drone features such as obstacle sensing and avoidance systems improve. Legislations on drones often have three main aims: (1) to regulate the use of airspaces; (2) to impose operational limitations; and (3) to outline administrative processes for permissions, licenses and authorizations [64]. The overall aim is therefore safety and security. The enactment of such rules is fundamental for further tapping into the potential benefits that drones come with in the various fields.

Several studies and reviews have been carried out in this area of regulations governing drone use and the implications of these on the industry. In the last year alone, Alamouri, Lampert [65] provided an overview of recent updates on drone regulations in the European Union and showed how regulations can help and hinder the use of the technology. Similarly, Hodgson and Sella-Villa [68] provide a review of the regulatory regimes in the United States, with particular reference to its application for academic research. They further highlight the complexities relating to restrictions on flying over critical infrastructure such as security installations when the locations of such facilities are classified for security reasons, as well as recommendations on how researchers can obtain exemptions from often sweeping restrictions. In the African context, Ayamga, Tekinerdogan [64] provide a review of the challenges that regulations pose for drone adoption and application, with specific focus on the field of agriculture. They argue that while the political commitment may be present in most Sub-Saharan African countries, regulations are often hampered by inadequate capacity to develop and enforce drone regulations.

## 5. Ethical and Safety Considerations

In terms of safety, drones are generally considered relatively safer than piloted aircraft for two main reasons; first, they are not piloted and so there is minimal risk of harm to the human controller in cases of a crash, and second, they do less damage on crashing because they are relatively smaller in size [69]. Modern drones also come with more safety features such as obstacle avoidance systems and return to launch buttons on controllers than their predecessors. This notwithstanding, drones come with some heightened concerns of safety due mainly also to their pilotlessness nature [70]. How safe a particular drone system would be is influenced, to some extent, by the drone configuration. For instance, rotary winged drones tend to fall stone-like in cases of rotor failure, while their fixed-wing counterparts tend to fall more gracefully. It is for this reason that drone licenses and flight permission are influenced by the type of drone.

Beyond safety is the ethical implications of research done using drones. The main issues of concern when discussing ethics in drone research are not markedly different from those that come up when using conventional research techniques like interviews, surveys, and FGDs, among others. Indeed, the key issues of privacy, confidentiality, and consent are still fundamental. The distinction of what constitutes the private sphere and public domain is critical. While drone data collection does not involve human test subjects *per se*, they often involve the observation of public places that humans are an intrinsic part of. In such contexts, it would be required that the data is recorded in a manner such that individuals are not personally identifiable, and if they were identifiable, disclosure of their identity outside of the research environment would not place them at the risk of any harm [71]. Studies that do not meet these thresholds may be subject to institutional ethics restrictions. On private property, however, studies will necessarily require consent from individuals to pass the ethics requirement. It is, for instance, not inconceivable that a drone captures an individual engaging in an illegal activity which would make them liable to criminal prosecution. The possibility of such accidental breaches of people's privacy means that drone operations over the private domain often requires researchers to obtain informed consent. Conversely, Sella-Villa [72] argues that drones are primarily data collection devices

whose impact on privacy is rather limited, as they are not substantially different from other camera-equipped technology. From this perspective, if a photograph is taken, the platform used is largely irrelevant. This notwithstanding, certain unique characteristics and qualities of drones means that their use as data collection tools in the social sciences brings to the fore key ethical concerns.

Issues regarding ethics in drone research, like in many other fields, is not a straight-forward one and is often riddled with inconsistencies and contradictions; what is private can quickly become public and vice-versa. For example, can people have private moments in a public park and how does one draw the distinction? There remain many grey areas and a lack of universality in principles regarding these requirements. For instance, what constitutes private information? While the airspace may reasonably be considered public space, would flying a drone over a farmer's field in open view require consent from them? What happens if they were growing marijuana on this field? Would institutional ethics review committees require researchers to gain informed consent for such drone operations? Moreover, obtaining informed consent from individuals in a study using drones can be a daunting task. This becomes impractical where, for instance this involves an indefinite number of people in a village.. In such a scenario, a community-wide forum prior to data collection becomes prudent. Through this, researchers could inform community members of what kind of information is to be collected and assure them of the protection of their anonymity, privacy and confidentiality [71]. This could engender public trust and buy-in on drone projects. This is especially critical in resource conservation in the interest of long-term sustainability of projects long after researchers leave research communities [69].

Despite their ubiquitous nature in the last few years, the capabilities of drones means that they are an excellent tool for surveillance, since they capture data from a vantage perspective inaccessible to other technologies [72]. There is, therefore, the need to be circumspect when applying them to data collection in the social sciences [70]. There is general agreement that researchers who employ drones to collect data should ideally submit their proposals to institutional review committees or some other oversight body for vetting to ensure compliance with ethical and regulatory standards [71]. While most drone studies, based on current ethical requirements, would qualify as exempt from such stringent reviews due to the minimal risk of harm to human subjects, researchers should nonetheless be aware of the possibility of ethical breaches that the collection of data in the public domain can occasion.

## 6. Conclusions

The last decade has seen a significant uptake in studies that use drones either as supporting tools or even as the primary methods of data collection. Drones can have important roles to play in mixed methods, especially in the areas of natural resource conservation, agriculture, tourism, among others. In this paper, we have discussed the increasingly widespread application of drones as tools for research in the social sciences. Given the unique capabilities of drones, there is the need for adequate ethical considerations when using them in research. While they hold enormous potential in multiple fields of study, certain fields such as geography and archaeology are already more inclined to their application than others. In archaeology, drones have enabled hitherto unobserved artifacts to become accessible to researchers. This has both positive and negative implications for heritage sites and indigenous populations. The application of drones in the fields of tourism studies and archaeology thus requires additional reflexivity to ensure that their use does not contribute to exploitation and looting of sites that were hitherto inaccessible. These considerations should fit into broader national guidelines and regulatory frameworks which should, in turn, be streamlined and be made less cumbersome to engender compliance. Certain barriers, which in most countries are financial, do not augur well for the adoption of the technology to reap the full benefits of their application.

Drones are already an inexorable part of society, and so spatial social scientists should be actively engaged with the use of this tool and be engaged in debates on the application

of the technology as a tool in their increasingly dynamic toolbox. This will ensure that the benefits inherent in the use of drones are maximised without exacerbating possibilities of breaches in ethics. A key plank of this engagement is the teaching of specialised courses in drones in institutions of higher learning. Such a drone course will not only help students acquire the technical skills to operate drones but also help them explore ways in which the tool can be applied in their own research specialisations, as well as enable them to engage critically with the ethical dilemmas inherent in its application in the social sciences. This latter discussion is critical as drones are becoming an integral tool in the toolbox of the social scientist as they become cheaper, safer, and more accessible.

**Author Contributions:** Conceptualization, O.H. and I.W.; methodology, O.H. and I.W.; formal analysis, O.H. and I.W.; resources, O.H.; data curation, I.W.; writing—original draft preparation, O.H. and I.W.; writing—review and editing, O.H. and I.W.; Project administration, O.H. Both authors read and agreed to the published version of the manuscript.

**Funding:** This research received no external funding, the APC was funded by Lund University.

**Institutional Review Board Statement:** Not applicable.

**Informed Consent Statement:** Not applicable.

**Conflicts of Interest:** The authors declare no conflict of interest.

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
