# Peer review of "The Use of Drones in the Spatial Social Sciences"

_drones, doi:10.3390/drones5040112_

Round 1

Reviewer 1 Report

Before starting my comments, I would like to congratulate the authors for having followed a clearer orientation in the work they intend to publish. It seems to me that this new version is more interesting and presents more relevant questions in the scientific domain and susceptible to discussion.

Positive aspects in this new version:

-The work is well written like the previous one;

-The organization of the chapters is equally clear;

-The goals are now very clear and noticeable;

-The bibliographical review seems to me to be adequate in all sections, but weaker in section 3;

-The subject deserves discussion and as such is subject to publication;

Aspects to improve:

- the discussion around drones is implicitly focused only on photographic cameras that these devices can carry, however a drone can carry other sensors without being just photographic; Therefore, it would make more sense to mention whenever they refer to a drone, they are referring to photographic images acquired by drones, and not exactly to the devices themselves!

- the parallelism between drones and GIS is not very acceptable, because drones are mentioned here as a data acquisition instrument with great spatial detail (data), and GIS technology allows, in its, sometimes, complex analytical process, to generate new information from the data integrated in the system database (generation new information from data). Data, by itself, may not constitute information, unless there is an organization of data layers and some spatial correlations, interrelations with other data are established, or specific analytical methods are applied, for instance. At most, a comparison could be made with other image acquisition platforms, but not with GIS!

- Section 3, dedicated to teaching, is perhaps the one that, in my opinion, is less well explored. Let's see, what is obtained, in the authors' approach, from drones are images, and the process of creating orthomosaics from images is what has been taught for years in photogrammetry. Whether the images are taken by aircraft or drones, the principle of creating orthomosaic is the same, what changes is the spatial resolution and consequently the level of detail. Photogrammetry has been taught in some areas of engineering and rigorously applied in various domains for the production of cartography at different scales! Although, enabling any common user (without proper training) to produce orthophotomaps from images acquired by drones is another thing, and debatable. But if this is the objective, the question always puts itself in the necessary fundamentals to do that and also the capacity to validate the cartographic producto generated by proper mathematical and statistical methods!

- Section 4 is perhaps the one that should be more debated at work, because it is the one where even more questions arise without clear answers. Security in the data acquisition and data use, ethical issues in the processing, use and availability of data… - should be discussed with specialists of various fields; - this work lacks that!

- In section 4, many of the issues raised are equally valid for any other image delivery platform (satellite, aerial photography or even any system for providing 360º video images of geographic space). What parallels should be done with other platforms? haven't these issues already been solved by big companies like Google, among others? Is it necessary to continue discussing these issues? so that, clear questions must be defined and concise answers must be presented.

Overall, it seems to me that the work presents the conditions to evolve, but it needs some improvements:

  1. Greater systematization of unsolved problems on data acquisition and data use;
  2. Presenting possible solutions to these problems, but substantiated after being evaluated by experts from various areas (air navigation, security, politics, etc.). Many of the problems are transversal to several different data acquisition technologies;
  3. Regarding the issue of teaching, I think it would be more appropriate to focus only on issues related to ethics in the use of data than on issues of operationalization of the drone, or on orthomosaic processing because this is already explored on photogrammetry;

In my opinion this work may be accepted, but it requires major changes.

Author Response

Dear Reviewer,

We would like to express our utmost appreciation for your detailed reading and useful comments and feedback. These have contributed to significantly improving our manuscript. We attach below our detailed point-by-point responses to specific comments.

Best regards,

Ola

Reviewer 2 Report

The authors discuss whether drones can be seen as a potential tool that could be used in social sciences. They highlighted this topic from three points of view:  teaching drones in higher educations, role of drone regulations, and ethical and political hurdles that may be caused by the use of drone, especially in social sciences.

Some issues to be respected:

  1. Uncrewed → it is better to replace it with "unmanned", where it is used moslty in the litratures, and also defined by EASA.
  2. Given drones’ advantage of providing access to the three-dimensional space → it is important to explain what does access means?
  3. In the section 2, the authors compared the use of satellite imagery to drone based one in the terms of social science: here I see somehow weakness, where you need to respect the application scale; while remote sensing has a potential in covering large scale areas, the drone based one is limited to non large scales.
  4. Section3: teaching drones should be re-introduced with following points of view:
    1. Teaching drones as hardware, here developing of drone systems, improvement of drone navigation and flight planning
    2. Teaching drone-based data capturing and data processing

Each scenario has its own requirements and imposes different teaching strategy. Now, we teach for e.g. drone-based data capturing and processing without needing to teach drone system concepts.

5. Section5:

The authors discussed the ethical and safety considerations that may affect the drone use, but they focus on the point if the drone has a payload like a camera! Does it differ if the drone does not have any camera, and used for social researches!

Author Response

(The authors gave the same response as above.)

Round 2

Reviewer 1 Report

I would like to thank the authors for taking some of my suggestions into account. I think the work reflects an important issue to be debated at the level of the civil community but also at the scientific and governmental level. I welcome the publication of this work and hope that the authors will contribute to the solution of some of the issues raised in this study.

Good work!